# The portfolio effect in a small-scale fishery reduces catch and fishing income variability in a highly dynamic ecosystem

**Andrés Vargas** [1] *, **Sebastián Restrepo** [2], **David Diaz** [1]

**1** Department of Economics, Universidad del Norte, Barranquilla, Atlántico, Colombia, **2** Departamento de Desarrollo Rural y Regional, Pontificia Universidad Javeriana, Bogotá, Colombia

* andresmv@uninorte.edu.co

**Data Availability Statement:** The minimal dataset and code are available without restriction from the

## Abstract

It is an increasingly accepted idea that biological diversity stabilizes ecosystem processes and the services they provide to society. By reducing biomass fluctuation, biodiversity could mitigate the impact of changing environmental conditions on rural incomes as long as people exploits a diverse set of natural assets. This effect is analogous to the risk-spreading function of financial portfolios. This paper presents evidence of the portfolio effect for an open-access artisanal fishery in an estuarine ecosystem, located in a Colombian Biosphere Reserve. Using catch statistics from 2002 to 2018, we evaluate the contribution of catch diversity to the stabilization of fishing income. We find that changes in catch composition are related to seasonal and interannual variations in salinity conditions. The portfolio effect arises from asynchronous fluctuations of fish species due to fluctuating environmental conditions. Catch diversification, instead of specialization, help achieve resilient fisheries.

## Introduction

The livelihoods of the rural poor are regularly dependent on natural resource use and the supply of ecosystem services. As a result, they are highly vulnerable to natural disaster shocks and environmental degradation, specially if their limited access to key markets, like credit and insurance, constrain their ability to maintain their consumption of goods and services through time. For them, the exploitation of open access resources is critical for meeting income and nutritional needs [1]. Consequently, they face high-income variability due to environmental and biological factors.

Strategies to reduce income risk include seasonal migration to areas with a greater fishing chance, gear diversification, collective action, additional livelihood strategies (e.g. farming, outside employment), and harvesting a portfolio of fish stocks [2–4]. Among these, catch diversification plays an important role when fish populations fluctuate asynchronously. Recent work on fisheries in North America has shown that catch diversification reduced variation in annual revenues [5–7], thus increasing economic stability, even during regime shifts [8].

The relationship between income variability and catch diversity stems from the role that biodiversity plays in the productivity and stability of natural systems [9, 10]. Three main

public repository https://github.com/andvarga-eco/portfolio_cgsm.

**Funding:** Andrés Vargas and David Diaz received funding from Universidad del Norte, www.uninorte.edu.co. Grant number 2017-30. The funder had no role in study design, data collection and analysis, decision to publish, or preparation of the manuscript.

**Competing interests:** The authors have declared that no competing interests exist.

mechanisms have been proposed to study the link between biodiversity and ecosystem function and services [11]: 1) complementary differences between species, 2) dominance by high-performing species, and 3) differential response of species to environmental conditions. The later of these, referred to as the insurance hypothesis, means that aggregate ecosystem properties vary less in more diverse communities [12]. For the well-being of society, the insurance hypothesis lends support to the claim that high diversity of response to environmental change among species is critical to the maintenance of valuable ecosystem services [13].

When limited to the time dimension, the insurance hypothesis is also known as the portfolio effect, which says that population diversity increases the temporal stability of a group of populations [12, 14]. In a multispecies fishery, it means that the variability of catch could be reduced if population densities of target species fluctuate asynchronously, allowing fishers to obtain a more stable income stream throughout the year. This effect is closely related to the literature in economics that studies how biodiversity insurances income against environmental shocks [15, 16].

The portfolio effect allow fishers to manage their exposure to changing environmental conditions through catch diversification. In other words, catch diversification is like having a portfolio of assets. According to modern portfolio theory in economics and finance, diversification is a way to allocate investment among alternative assets, to obtain the higher expected return for a given level of risk. In this sense, the expected yield and variance of a fisher's portfolio depends on species' response to variations in their environment [17]. Moreover, since species differ in their response to environmental variations, then environmental risk is diversifiable, i.e. it could be mitigated through diversification.

However, in contrast with a financial investor, a fisher in a multispecies fishery cannot fully decide the weights each species has on his/her portfolio. Rather, portfolio diversification is attained through fishing strategies and gear of choice. Using selective gear implies a less diversified catch than using a non-selective gear. Catch composition is thus a function of gear, strategies, and environmental conditions. Insofar as fishers' do not fully control their portfolio composition, then, the idea that they can construct diversified portfolios to achieve attractive returns for a given level of risk, as postulated by modern portfolio theory, is of little use. Rather, it seems better to assume that if environmental risks are of primary concern, then success depends on adopting strategies, e.g. non-selective gear, that allow fishers to maintain their catch and income no matter what the environment is like.

Most of the studies on the stabilizing effect of diversification have occurred in the context of highly regulated commercial fisheries in developed countries. But, in unregulated tropical fisheries, catch diversification effects are understudied. The scant evidence shows that diversification has contributed to insulate fishers against long-term declines in catch rates [18]. In this study, we test the stabilizing effect of catch diversification, i.e. portfolio effect, in the unregulated open access multispecies fishery of the Ciénaga Grande de Santa Marta (CGSM), the largest coastal lagoon in Colombia [19]. Our study site is ideal for studying the portfolio effect, since it is an estuarine ecosystem in which fish assemblages change due to variations in environmental conditions, specially salinity [20, 21].

Here, we focus on changes in salinity levels because of their influence on the richness and abundance of fish species available to fishers, thus affecting the composition and volume of harvested fish for a given fishing effort. Therefore, we analyse the relationship between catch and salinity to test the portfolio effect. The fishers we study use only one type of gear (cast net), a non-motor boat, and are partially isolated. For them, fish diversity has a significant role in stabilizing income, given their limited access to credit, capital, and labour markets. Census data from 2018 shows that about 65% of the population in the area is considered poor [19]. Also, data from a nearby fishing community shows that less than 2% of households requested

loans from the formal financial sector, and that multispecies fishing is key to diversifying risk [22].

## Materials and methods

### Study site

The CGSM is a delta-lagoon complex composed of water bodies with estuarine behavior. Paleochannels connect them with the Magdalena River, swamps, and extensive mangrove forests. It has two central water bodies: the Ciénaga Grande coastal lagoon (450 $km^2$) and the Ciénaga de Pajarales (120 $km^2$), around 43% of the delta complex area [23]. The CGSM complex is connected to the north with the Caribbean sea; to the east with the Sierra Nevada de Santa Marta and its downstream rivers, which produce intense discharges; and to the west with the Magdalena River. The seawater-freshwater interaction throughout the year greatly influences the physicochemical features of the wetland, in particular its salinity levels (Fig 1), whereas interannual variability in salinity is related to the phases of El Niño Southern Oscillation [24]. Changes which in turn affect the fishery [25].

The artisanal fishery in the CGSM is one of the largest in Colombia, with around 3500 fishers operating every day under an open-access regime. The fishery is *de facto* unmanaged, but routinely monitored. Most of the fishers use canoes, two fishermen per canoe, and a cast net as the gear of choice. They target different fish groups, such as gerreids, ariids, and mugilids [23, 26, 27]. During the last three decades, many fishing cooperatives and associations have been created with the aim of moving towards the management of fishing activities in the CGSM

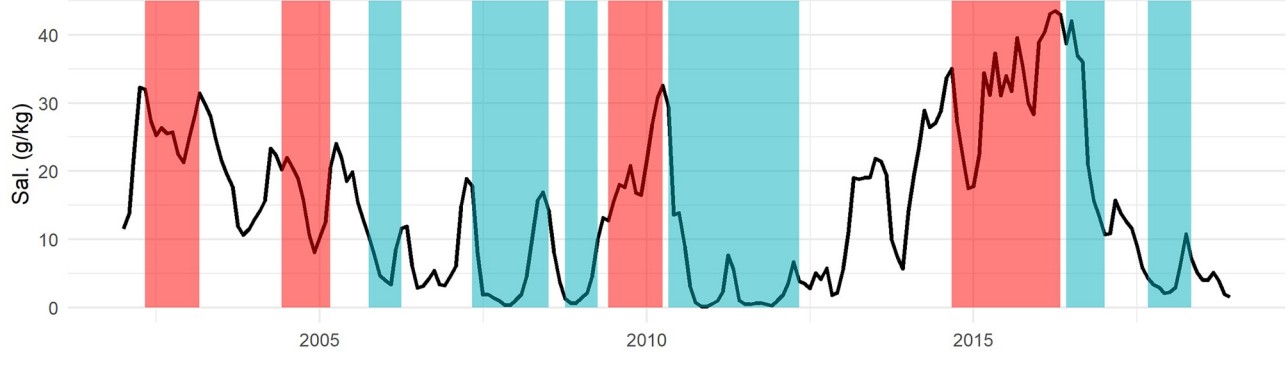

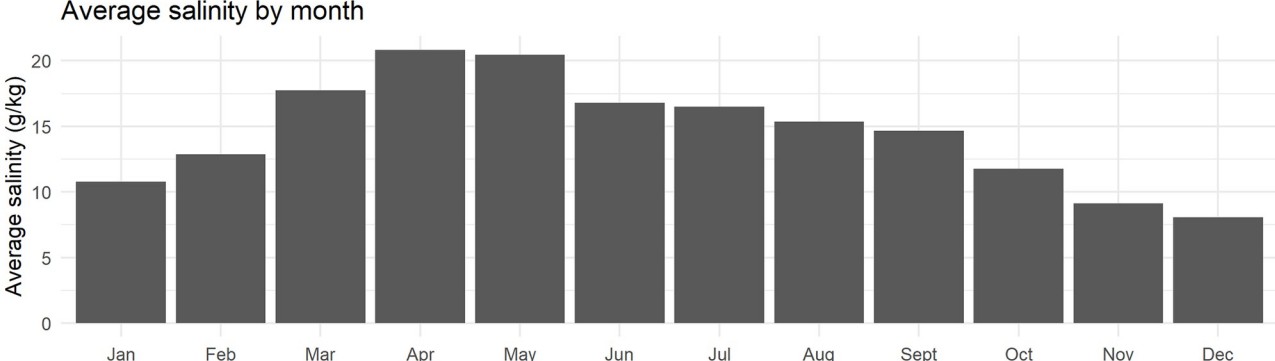

**Fig 1. Salinity: Interannual and seasonal variability.** Red: El Niño. Blue: La Niña.

[27]. Two large associations work as cooperative federations that group about 40 of the 69 existing cooperatives.

## Data

The data was obtained from the system of information on fisheries, Sipein, maintained by Invemar, Colombia's coastal and marine research institute (sipein.invemar.org.co/informes/tallas/externos/ind/). Sipein's database includes information on catch (kg), effort (number of trips per month), and income (Colombian pesos). Catch and income data are broken down by species, fishing method, and landing port, whereas effort data is disaggregated at the fishing gear and landing port levels. Although it is reported that more than 100 species are commercially exploited, seven species account for two thirds or more of the catch [26]. The data contains information for the seven most commercialized species, while remaining species are grouped into the category "other". Target species differ in their preferred salinity habitat (Table 1).

For this study, we use data for the casting net fishing method (*atarraya*) and the landing port Nueva Venecia. Two reasons justify this decision: First, the casting net, a cheap method with little capital and labor requirements, widely used in the area, allows fishers to catch several species year round. Although their main target is *Mugil incilis*, the non-selective character of the gear allows them to get a valuable bycatch. Second, most of the harvest traded in the landing port of Nueva Venecia comes from fishers belonging to the fishing communities of Nueva Venecia and Buenavista, which are isolated stilt villages located in the middle of the Ciénaga de Pajaral, about an hour's boat ride from the closest town. These characteristics mean that fishers' main mechanism for coping with environmental variability is their access to a diverse fishing portfolio, rather than seasonal migration or outside employment.

Water salinity is characterized using data from the Ciénaga Grande de Santa Marta Monitoring program, which is carried out by Invemar (www.invemar.org.co/inf-cgsm). Salinity (g/kg) is measured at 14 point locations each month. The salinity variable we use is the average of the measures taken at those points. Our data sample goes from January 2002 to December 2018.

## Analytical methods

To evaluate whether catch diversification reduces catch and income variability, we look for evidence of asynchronous fluctuations of species, community-wide synchrony, and analyse the relationship between salinity and catch composition. Asynchronous fluctuation, and a

**Table 1. Main species caught and salinity habitat.**

| Specie | Freshwater | Brackish | Marine |
|---|---|---|---|
| *Ariopsis canteri* | ▓ | ▓ | |
| *Mugil incilis* | ▓ | | ▓ |
| *Elops smithi* | | ▓ | ▓ |
| *Cathorops mapale* | ▓ | ▓ | ▓ |
| *Oreochromis nicolitus* | ▓ | ▓ | |
| *Eugerres plumieri* | ▓ | ▓ | |
| *Megalops atlanticus* | ▓ | ▓ | ▓ |

Source: Robertson y Van Tassell, 2019, Shorefishes of the Great Caribbean (https://biogeodb.stri.si.edu/caribbean/es/pages)

differential response to changes in salinity conditions, indicate that diversification has allowed fishers to mitigate environmental risk.

**Community-wide synchrony.** According to the portfolio effect, asynchronous fluctuations of species increase the temporal stability of community level variables. To relate catch to species population dynamics, we use a simple multispecies fishery model. Let $h_{i,t}$ denote the catch for species $i$ and $p_{i,t}$, its selling price, then total catch in period $t$ is given by $H_t = \sum_{i=1}^{s} h_{i,t}$, whereas total revenue $Y_t = \sum_{i=1}^{s} p_{i,t} h_{i,t}$. Total catch is characterized by a Schaefer equation

$$H_{t+1} = (\sum_{i=1}^{s} q_i X_{i,t+1}) E_{t+1} \tag{1}$$

where $q_i$ is the catchability coefficient, $X_{i,\,t+1}$ is stock size, and $E_{t+1}$ is the aggregate effort level. Therefore, total catch per unit of effort, $CPUE_{t+1}$, is defined as

$$CPUE_{t+1} = \sum_{i=1}^{s} q_i X_{i,t+1} \tag{2}$$

From Eq (1), it is apparent that the temporal variability of $H_t$ is driven by the dynamics of the aggregate effort and each species population, whereas, for $CPUE_t$, the sole driver is population dynamics. For total revenue and revenue per unit of effort, $YPUE_t = Y_t/E_t$, price dynamics is also important.

In a community, the temporal variance of a community-level variable is directly related to the synchrony of the variables at the species level, which, in a fluctuating environment, depends on the species' response to environmental fluctuations. Accordingly, the temporal variance for $CPUE$ can be expressed as

$$\text{Var}(CPUE) = \sum_i q_i^2 Var(X_i) + \sum_{i<j} q_i q_j Cov(X_i, X_j) \tag{3}$$

Since we do not observe $X_i$, we define $cpue_i = h_i/E$ and, therefore $CPUE_t = \Sigma_i cpue_i$, thus

$$\text{Var}(CPUE) = \sum_i Var(cpue_i) + \sum_{i<j} Cov(cpue_i, cpue_j) \tag{4}$$

From the previous, it is clear that negatively correlated species reduces the variance of $CPUE$. Since there are abiotic forces that contributes to the synchronization of population dynamics, we assess the degree of community wide synchrony using the following metrics

$$\phi = \frac{\sigma_Z^2}{\left(\sum_i \sigma_{zi}\right)^2} \tag{5}$$

$$\eta = \frac{1}{s} \sum_i corr\left(z_i, \sum_{j \neq i} z_j\right) \tag{6}$$

The statistic $\phi$ [28] is the ratio of observed variance in $Z$ and the maximum possible variance that would arise if all components of $Z$ were perfectly correlated. It takes values between 0, perfect asynchrony, and 1, perfect synchrony. The metric $\eta$ [29] is the average across species of the correlation between catch per unit of the aggregate effort of each species and the total catch per unit effort of all other species in the group. This metric takes a minimum value of −1, perfect asynchrony, and a maximum of + 1 if species are perfectly synchronized. And

advantage of $\eta$ is that it is centred at 0 when species fluctuate independently. These statistics are also computed for revenue per unit of effort, total catch, and total revenue. $Z$ stands for any temporal community-level variable and $z_i$ the equivalent temporal variable for species $i$. We calculate the metrics for *CPUE*, *YPUE*, *H*, and *Y*.

**Salinity and catch composition.** While the aforementioned statistics allows us to determine the degree of synchronization, we cannot tell how each species responds to changes in salinity levels and, thus, their contribution to the stabilization of *CPUE*. Using a linear stock-effort relationship [30, 31], species $i$ abundance is assumed to evolve according to Eq (7)

$$X_{i,t+1} = \theta_{0i} + \theta_{1i}X_{i,t} + \theta_{2i}E_t + f_i(S_t) \tag{7}$$

where $f_i(S_t)$ represents the impact of salinity, $S_t$, on current adult abundance. In a source-sink system like the one we are studying, the abundance of freshwater species, such as *Oreochromis nicolitus*, depends on the connection with the Magdalena River, whereas other species, like *Megalops atlanticus* and *Ariopsis canteri*, can move between the Caribbean sea and the lagoons. For each species $i$, we can write catch per unit of aggregate effort for each species as $CPUE_{it} = q_i X_{it}$, and using Eqs (1) and (7)

$$CPUE_{i,t+1} = \gamma_{0i} + \gamma_{1i}CPUE_{it} + \gamma_{2i}E_t + \gamma_{3i}S_{t+1} + \gamma_{4i}S_{t+1}^2 + \upsilon_{i,t+1} \tag{8}$$

Where the errors $\upsilon_{it}$, capture unobserved variables and random shocks. Because $i = 1, 2,.., 8$ we have a system of eight equations. The regressors for each equation are predetermined or exogenous, thus, each equation may be estimated independently by OLS. However, since species may be impacted by unobserved physical or environmental variables, then, the errors, $\upsilon_{it}$, are likely to be related. We perform a Breusch-Pagan test of independence of equations to test the hypothesis of cross-equation error independence. We reject the hypothesis, therefore, we estimate the parameters of Eq (8) for each species $i$ using the seemingly unrelated regressions method, SUR.

## Results

Catch composition changed along a salinity gradient (Fig 2). In oligo-haline conditions, 0-5 g kg$^{-1}$, freshwater species like *Oreochromis nicolitus* have a greater presence than esturaine especies, *Ariopsis canteri, Cathorops mapale, Elops schmithi*. At intermediate levels, 6 to 30 g kg$^{-1}$ these estuarine species, as well as *Megalops atlanticus*, account for a greater share of total catch, whereas at levels above 30 g kg$^{-1}$, *Eugerres plumieri* represents about 70% of total catch. This last specie is the most important at all salinity levels, which is in accordance with its capacity to occupy waters with an ample salinity range. These changes in composition are coherent with the salinity habitat identified for each specie (Table 1).

Mean values for catch per unit effort, CPUE, are larger at intermediate salinity levels, 36.5 and 39.9 kg for ranges 6-18 and 19-30 g kg$^{-1}$, whereas for 0-5 and >30 g kg$^{-1}$ are 29.7 and 35.2 kg. For revenue per unit effort, YPUE, mean values are higher for salinity levels above 18 g kg$^{-1}$ (Fig 3). Neither CPUE nor YPUE show a particular time trend.

Synchrony metrics (Table 2) for total catch, H, and income, Y, are higher than for CPUE and YPUE, which confirms that aggregate effort, $E$, contributes to the synchronization of H and Y. Once the effect of the aggregate effort is taken into account, it is clear that asynchronous variation among species caused a reduction in the variance of CPUE, $\varphi = 0.173$ and $\eta = -0.054$. Interestingly, lower values for YPUE, $\varphi = 0.095$ and $\eta = -0.199$, indicate that prices also varied asynchronously among species, further stabilizing income.

Results from the SUR model (Table 3) show that the relationship between salinity and catch varies across species. For one specie, *Mugil incilis*, the relationship is positive. Three species

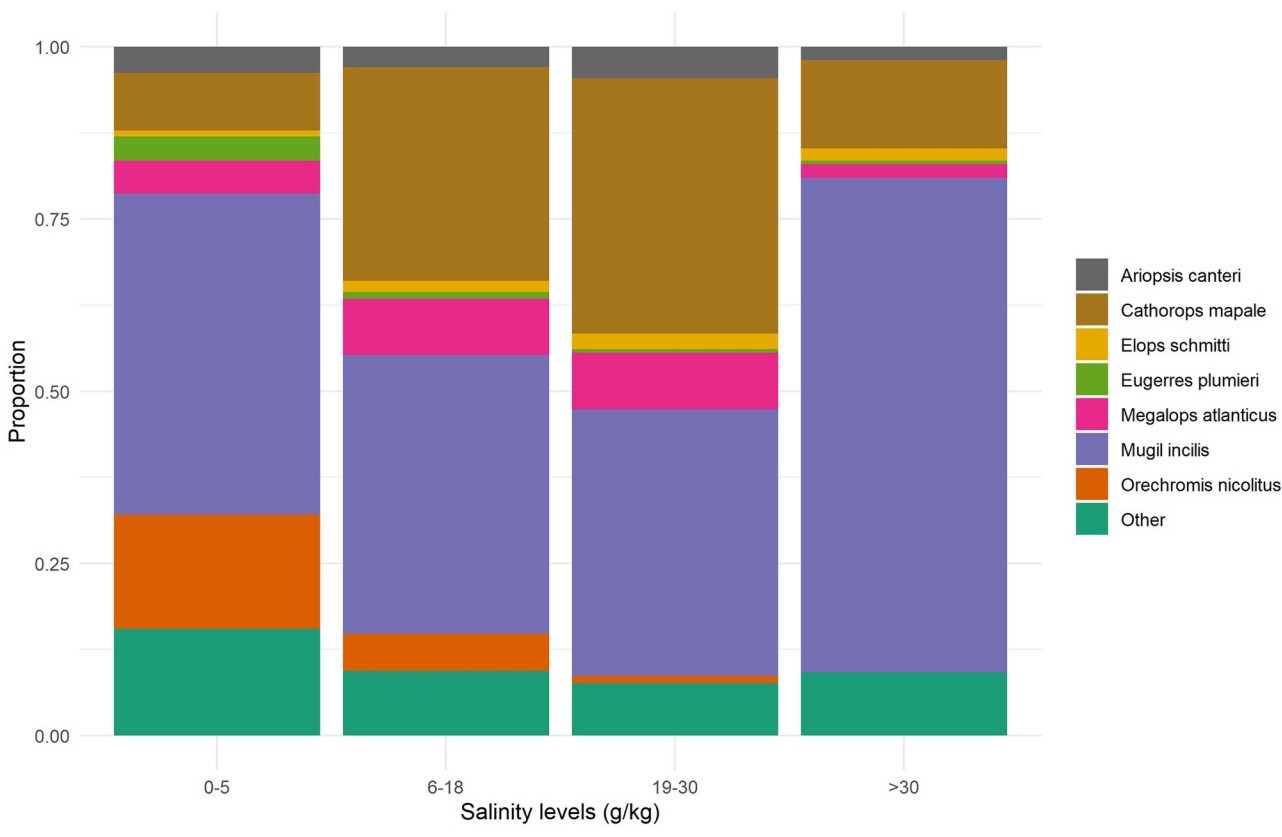

**Fig 2. Catch composition and salinity, 2002-2018.** Landing port: Nueva Venecia

(*Ariopsis canteri, Elops schimithi, Cathorops mapale*) exhibit an inverted u pattern, with a turning point between 19 and 22.6 g kg$^{-1}$. For remaining species, increases in salinity levels cause reductions in catch. Using $\hat{\gamma}_{3i}$ and $\hat{\gamma}_{4i}$ point estimates, the marginal effect of salinity levels on *CPUE* can be obtained as $\sum_i \hat{\gamma}_{3i} + 2S\sum_i \hat{\gamma}_{4i}$. This effect is positive, but there is a decreasing rate for salinity levels in the range 0-23 g kg$^{-1}$, and negative for salinity values greater than 23 g kg$^{-1}$, which is coherent with the estuarine characteristics of the ecosystem.

## Discussion

By diversifying their catch, fishers manage their exposure to environmental risk, helping them stabilize catch rates and income. Our results indicate that stabilization is thanks to the differential response of target species to salinity conditions, which produces asynchronous fluctuations of fish populations. In an estuarine ecosystem, where salinity is the most important factor regulating the temporal patterns in the diversity and abundance of fish, the portfolio effect is driven by the salinity tolerance of fish species. More precisely, two effects seem to be at play. First, the main targeted species (*Mugil incilis*) has a wide range of salinity tolerance and is also the most important in volume; thus community stability could be enhanced due to the dominance of this species. Second, different species or traits are favoured under various salinity regimes.

Synchrony statistics show that the stabilization effect is greater for income than for catch rates. Since the former is the latter multiplied by selling prices, the evidence then suggests that

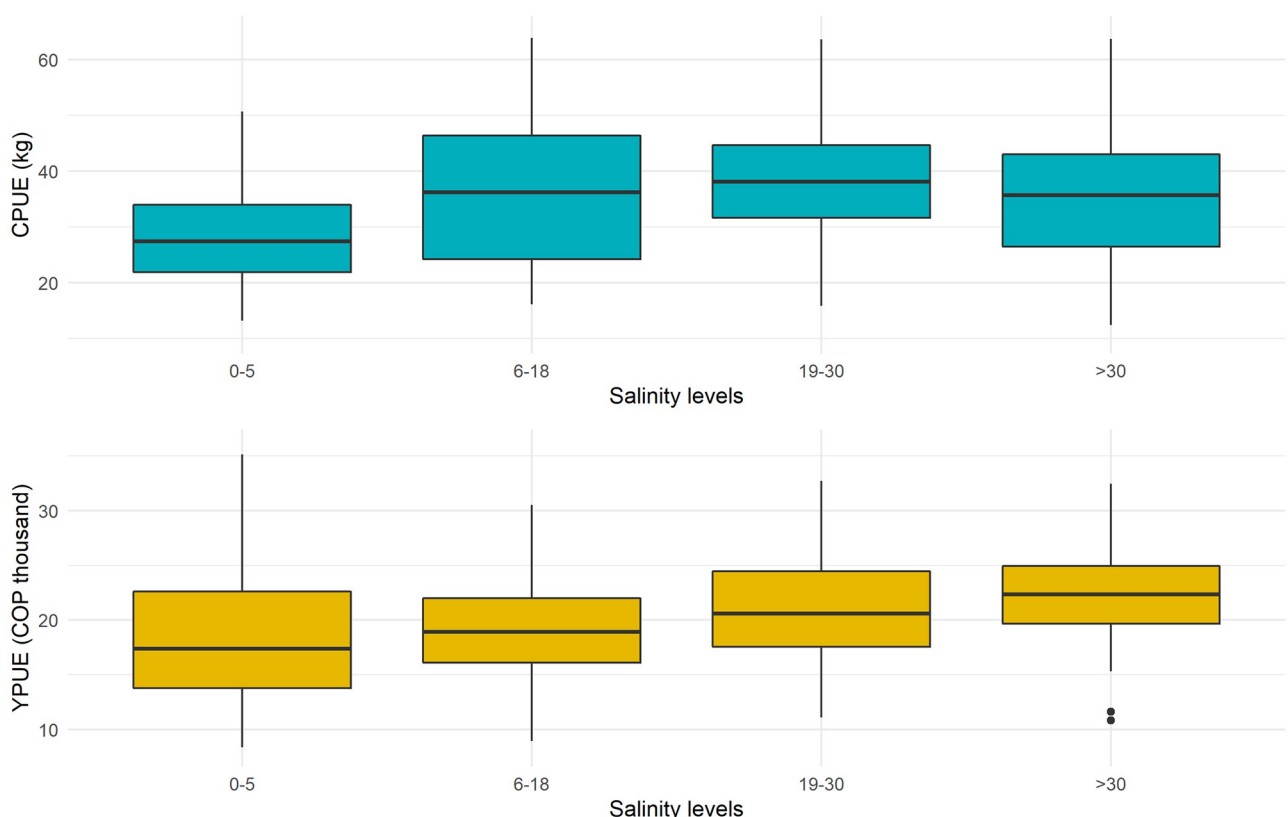

**Fig 3. Catch and income per unit of effort, 2002-2018.**

**Table 2. Synchrony statistics.**

|  | H | Y | CPUE | YPUE |
|---|---|---|---|---|
| $\phi$ | 0.411 | 0.375 | 0.173 | 0.095 |
| $\eta$ | 0.384 | 0.298 | -0.054 | -0.199 |

**Table 3. Effect of salinity on catch composition.**

| Variables | cpue1 | cpue2 | cpue3 | cpue4 | cpue5 | cpue6 | cpue7 | cpue8 |
|---|---|---|---|---|---|---|---|---|
| cpuei (t-1) | 0.684 | 0.597 | 0.502 | 0.766 | 0.684 | 0.800 | 0.659 | 0.58 |
|  | (0.051) | (0.055) | (0.056) | (0.046) | (0.051) | (0.045) | (0.064) | (0.065) |
| L(t-1) | -0.00018 | -0.0004 | 0.00007 | 0.0004 | 0.00005 | 0.00004 | -0.0008 | -0.0001 |
|  | (0.00008) | 1(0.0007) | (0.00003) | (0.0005) | (0.0003) | (0.00006) | (0.0002) | (0.0002) |
| Salinity | 0.038 | 0.11 | 0.025 | 0.39 | -0.058 | -0.0064 | -0.028 | -0.023 |
|  | (0.018) | (0.054) | (0.008) | (0.112) | (0.024) | (0.0054) | (0.015) | (0.016) |
| sal2 | -0.0009 |  | -0.0006 | -0.0089 |  |  |  |  |
|  | (0.0004) |  | (0.00019) | (0.0027) |  |  |  |  |
| Constant | 0.212 | 10.80 | -0.11 | -1.65 | 0.952 | -0.074 | -0.46 | 2.31 |
|  | (0.407) | (3.45) | (0.177) | (2.47) | (1.54) | (0.346) | (0.93) | (1.07) |
| R-squared | 0.61 | 0.43 | 0.53 | 0.74 | 0.62 | 0.67 | 0.54 | 0.41 |

SUR model. Breusch-Pagan test of independence:chi2(28)=69.276. Standard errors in parentheses. Seasonal dummies included. i:1 Ariopsis canteri, i:2 Mugil incilis, i3: Elops schimitti, i:4 Cathorops mapale, i:5 Oreochromis nicolitus, i:6 Eugerres plumieri, i: 7 Megalops atlanticus, i: 8 Other

prices also tend to move asynchronously, further amplifying the portfolio effect. This is an important aspect that deserves further attention.

The portfolio effect in ecology has been connected to ideas of portfolio management in economics and finance, in particular to those based on the mean-variance trade-off of modern portfolio theory [12]. Generally, the modern portfolio theory builds a framework for the selection of investment portfolios that maximize the return for a given level of risk, or one that attain a desired level of return at a minimum risk. To implement the method, investors need to calculate the returns, variances, and covariances of assets. Based on this information, they decide how much of their wealth is allocated to each asset.

Although this logic could be useful for implementing an ecosystem-based fishery management approach in a regulated fishery [32], it is of limited use in an artisanal multispecies fishery where fishers cannot completely determine how much of their effort to allocate to each species. Although artisanal fishers make strategic decisions in order to influence their catch, they do not have the degree of control in catch allocation decisions required to implement a mean-variance portfolio strategy. Not to mention, the need to compute the required statistics. In this sense, it is better to view the idea of a portfolio as a diversification strategy devised through experience, and which has proven to be effective in allowing generations of fishers to maintain their livelihoods in a changing environment. Our results coincide with recent evidence showing that, in the tropics, indigenous communities tend to adopt highly diversified agricultural strategies [33], or that artisanal fishers prefer fishing gear that captures a great diversity of species [22].

Our results highlight the importance of biodiversity for human well-being in general, and for maintaining livelihoods in rural settings in particular [17]. The evidence that diversification helps to stabilize catch and income is in line with agricultural development policies that focus on the promotion of crop diversity rather than specialization [34]. This, in stark contrast to recent interventions in this area of study, which have promoted fish farming projects specialized in one species (*Megalops atlanticus*). Development projects that consider the stabilizing effect of diversification are more likely to benefit households, whereas those focused on maximizing revenues through specialization are prone to fail due to their greater vulnerability to environmental change.

Our research provides evidence in support of policies that take advantage of functional diversity to help fishers to adapt to environmental variability through diversification, while, at the same time, contributing to enhance and protect ecosystem services provisioning.

## Author Contributions

**Conceptualization:** Andrés Vargas, Sebastián Restrepo, David Diaz.

**Data curation:** Andrés Vargas.

**Formal analysis:** Andrés Vargas.

**Funding acquisition:** Andrés Vargas, David Diaz.

**Investigation:** Sebastián Restrepo.

**Methodology:** Andrés Vargas, Sebastián Restrepo.

**Project administration:** Andrés Vargas.

**Writing – original draft:** Andrés Vargas.

**Writing – review & editing:** Andrés Vargas, Sebastián Restrepo, David Diaz.

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
