## [Decision Letter · Decision Letter 0]

25 Feb 2022

PONE-D-22-02234

The portfolio effect in a small-scale fishery reduces catch and fishing income variability in a highly dynamic ecosystem

PLOS ONE

Dear Dr. Vargas,

Thank you for submitting your manuscript to PLOS ONE. After careful consideration, we feel that it has merit but does not fully meet PLOS ONE’s publication criteria as it currently stands. Therefore, we invite you to submit a revised version of the manuscript that addresses the points raised during the review process.

The manuscript deals with a very interesting and timing issue, and I tend to agree with the comments/suggestions made by reviewer #1. The methodology used in the analysis is robust and sound and the metric used in relation to the paper by Gross K., et al. 2014 is a good choice. I suggest that in your revision you pay also attention to the use of English and improve the text in order to communicate more clearly your message.

We look forward to receiving your revised manuscript.

Kind regards,

Andrea Belgrano, Ph.D.

Academic Editor

PLOS ONE

Journal Requirements:

2. Thank you for stating the following financial disclosure: "Andrés Vargas and David Diaz received funding from Universidad del Norte, www.uninorte.edu.co. Grant number 2017-30. The funder had no role in study design, data collection and analysis, decision to publish, or preparation of the manuscript."

We note that one or more of the authors is affiliated with the funding organization, indicating the funder may have had some role in the design, data collection, analysis or preparation of your manuscript for publication; in other words, the funder played an indirect role through the participation of the co-authors. If the funding organization did not play a role in the study design, data collection and analysis, decision to publish, or preparation of the manuscript and only provided financial support in the form of authors' salaries and/or research materials, please do the following:

a. Review your statements relating to the author contributions, and ensure you have specifically and accurately indicated the role(s) that these authors had in your study. These amendments should be made in the online form.

b. Confirm in your cover letter that you agree with the following statement, and we will change the online submission form on your behalf: 

“The funder provided support in the form of salaries for authors [insert relevant initials], but did not have any additional role in the study design, data collection and analysis, decision to publish, or preparation of the manuscript. The specific roles of these authors are articulated in the ‘author contributions’ section.

4. Please upload a new copy of Figures 1, 2, 3, and 4 as the detail is not clear. Please follow the link for more information: " ext-link-type="uri" xlink:type="simple">https://blogs.plos.org/plos/2019/06/looking-good-tips-for-creating-your-plos-figures-graphics/"
" ext-link-type="uri" xlink:type="simple">https://blogs.plos.org/plos/2019/06/looking-good-tips-for-creating-your-plos-figures-graphics/"

5. Please ensure that you refer to Figures 1, 2, 3, and 4 in your text as, if accepted, production will need this reference to link the reader to the figure.

6. We note that Figure 1 in your submission contain map images which may be copyrighted. All PLOS content is published under the Creative Commons Attribution License (CC BY 4.0), which means that the manuscript, images, and Supporting Information files will be freely available online, and any third party is permitted to access, download, copy, distribute, and use these materials in any way, even commercially, with proper attribution. For these reasons, we cannot publish previously copyrighted maps or satellite images created using proprietary data, such as Google software (Google Maps, Street View, and Earth). For more information, see our copyright guidelines: http://journals.plos.org/plosone/s/licenses-and-copyright.

“I request permission for the open-access journal PLOS ONE to publish XXX under the Creative Commons Attribution License (CCAL) CC BY 4.0 (http://creativecommons.org/licenses/by/4.0/). Please be aware that this license allows unrestricted use and distribution, even commercially, by third parties. Please reply and provide explicit written permission to publish XXX under a CC BY license and complete the attached form.

7. We note you have included a table to which you do not refer in the text of your manuscript. Please ensure that you refer to Table 2 in your text; if accepted, production will need this reference to link the reader to the Table.

Reviewers' comments:

Reviewer's Responses to Questions

**Comments to the Author**

1. Is the manuscript technically sound, and do the data support the conclusions?

Reviewer #1: Yes

2. Has the statistical analysis been performed appropriately and rigorously? 

Reviewer #1: I Don't Know

3. Have the authors made all data underlying the findings in their manuscript fully available?

Reviewer #1: Yes

4. Is the manuscript presented in an intelligible fashion and written in standard English?

Reviewer #1: No

5. Review Comments to the Author

Reviewer #1: Dear authors,

This is a really interesting and relevant manuscript that address and important question: how do secure access to resources and stable incomes in a dynamic environment and an open access regime. I found the manuscript really interesting, but would also encourage you to work further on communicating your message. The introduction is somewhat confusing and can be clarified, complicated words could be more clearly explained, the links between biodiversity, ecosystem function and resilience can be better referenced, and the text could benefit from a professional language editor (and English is not my first language). Some references to the figures were missing, the reference list is incomplete, and I was not able to follow all aspects of the modeling. If you are able to address these issues, this will be a really important and nice paper. Please find my detailed comments below:

1. Introduction:

a) "specially if their capacity to smooth consumption through access to credit..." this is not very clear and can probably be reworded.

b) When you speak about artisanal fisheries, you repeat what you have said in the previous section. Perhaps delete the first sentence of the second paragraph, and instead refer to "Strategies to recuse income risks among artisanal fisheries include....."

c) do you really need the word "asynchronously" here and in the rest of the ms? Isn't it enough with just "fluctuate"?

d) please explain more clearly the links between ecosystem diversity and function. This is a long-standing debate with lots of opinions. You can elaborate and provide further references. This is a favorite from my perspective, but it is very old and probably lots of better and newer things out there (Elmqvist et al. 2003): https://esajournals.onlinelibrary.wiley.com/doi/pdf/10.1890/1540-9295(2003)001[0488:RDECAR]2.0.CO;2 and hopefully relevant to your study.

e) on line 44-45 you refer to survival. Is that correct?

f) the fishers in the study: do you have any data that illustrates your points of how diversity stabilize income, and that they do not have access to credit, etc (last section of introduction)?

Materials and methods

a) please check that all figures are referred to properly. All references to figures are "Fig ??" in the pdf I have access to (maybe this is just a formatting issue).

b) line 114 "this effect" - please clarify what this refers to - it is unclear.

c) Line 17, refers to "methods" but this is consuming since it is already part or the methods section. Consider revising to "Analytical method" or similar.

d) Line 124 "the the" - delete one "the"

I had a hard time understanding the SUR model (and they results from it), but maybe that is just because it is outside of my area of expertise.

Discussion

a) maybe explain the mean-variance trade off for the uninformed reader?

b) please consider if you want to include this discussion about functional diversity etc., and if so, include more references.

References:

Some references have full first names (e.g., 1, 30, 31, 32) but most don't. Please check the formatting. Reference 28 is missing a year, and 29 is missing a capital letter for the first name "Andersson S." Check all formatting and consistency with PLOS one requirements.

6. PLOS authors have the option to publish the peer review history of their article (what does this mean?). If published, this will include your full peer review and any attached files.

Reviewer #1: No

---

## [Author Response · Author response to Decision Letter 0]

22 Jun 2022

Reviewer #1

1. Introduction:

a) "specially if their capacity to smooth consumption through access to credit..." this is not very clear and can probably be reworded.

R/ It now reads

“…specially if their limited access to key markets, like credit and insurance, constrain their ability to maintain their consumption of goods and services through time”

b) When you speak about artisanal fisheries, you repeat what you have said in the previous section. Perhaps delete the first sentence of the second paragraph, and instead refer to "Strategies to recuse income risks among artisanal fisheries include....."

R/ Changed as suggested

c) do you really need the word "asynchronously" here and in the rest of the ms? Isn't it enough with just "fluctuate"?

R/ Yes, since the degree of species synchrony is key to the stability of the species community

d) please explain more clearly the links between ecosystem diversity and function. This is a long-standing debate with lots of opinions. You can elaborate and provide further references. This is a favorite from my perspective, but it is very old and probably lots of better and newer things out there (Elmqvist et al. 2003): https://esajournals.onlinelibrary.wiley.com/doi/pdf/10.1890/1540-9295(2003)001[0488:RDECAR]2.0.CO;2 and hopefully relevant to your study.

R/ Thanks for the suggestion. A brief explanation on the relationship between biodiversity and ecosystem function and services is added, including two references: Elmqvist et al. (2003) and Balvanera et al. (2015). 

“The relationship between income variability and catch diversity stems from the role that biodiversity plays in the productivity and stability of natural systems [9, 10]. Three main mechanisms have been proposed to study the link between biodiversity and ecosystem function and services [11]: 1) complementary differences between species, 2) dominance by high-performing species, and 3) differential response of species to environmental conditions. The later of these, referred to as the insurance hypothesis, means that aggregate ecosystem properties vary less in more diverse communities [12]. For the well-being of society, the insurance hypothesis lends support to the claim that high diversity of response to environmental change among species is critical to the maintenance of valuable ecosystem services [13]”.

e) on line 44-45 you refer to survival. Is that correct?

R/ Changed to success

f) the fishers in the study: do you have any data that illustrates your points of how diversity stabilize income, and that they do not have access to credit, etc (last section of introduction)?

R/ The assertion is based on our observations during fieldwork in the area. To better illustrate it, we added statistical data concerning poverty in the area, as well as evidence from a nearby fishing community showing their limited access to credit. 

2. Materials and methods

a) please check that all figures are referred to properly. All references to figures are "Fig ??" in the pdf I have access to (maybe this is just a formatting issue).

R/ Checked

b) line 114 "this effect" - please clarify what this refers to - it is unclear.

R/ The paragraph was placed in the wrong place. It was deleted

c) Line 17, refers to "methods" but this is consuming since it is already part or the methods section. Consider revising to "Analytical method" or similar.

R/ Changed as suggested

d) Line 124 "the the" - delete one "the"

R/ Changed as suggested

I had a hard time understanding the SUR model (and they results from it), but maybe that is just because it is outside of my area of expertise.

R/ We reworded the paragraph below equation (8) to better explain the model. 

Discussion

a) maybe explain the mean-variance trade off for the uninformed reader?

R/ An explanation was added as suggested

“Generally, the modern portfolio theory builds a framework for the selection of investment portfolios that maximize the return for a given level of risk, or one that attain a desired level of return at a minimum risk. To implement the method, investors need to calculate the returns, variances, and covariances of assets. Based on this information, they decide how much of their wealth is allocated to each asset.”

b) please consider if you want to include this discussion about functional diversity etc., and if so, include more references.

R/ We decided not to include the discussion

References:

Some references have full first names (e.g., 1, 30, 31, 32) but most don't. Please check the formatting. Reference 28 is missing a year, and 29 is missing a capital letter for the first name "Andersson S." Check all formatting and consistency with PLOS one requirements.

R/ List of references checked and formatted in accordance with PLOS requirements

---

## [Editor Report · Decision Letter 1]

27 Jun 2022

The portfolio effect in a small-scale fishery reduces catch and fishing income variability in a highly dynamic ecosystem

PONE-D-22-02234R1

Dear Dr. Vargas,

We’re pleased to inform you that your manuscript has been judged scientifically suitable for publication and will be formally accepted for publication once it meets all outstanding technical requirements.

Kind regards,

Andrea Belgrano, Ph.D.

Academic Editor

PLOS ONE

Additional Editor Comments (optional):

Thank you for addressing all the comments/suggestions made during the review process, the revised manuscript reads very well and with clarity.

---

## [Editor Report · Acceptance letter]

29 Jul 2022

PONE-D-22-02234R1 

The portfolio effect in a small-scale fishery reduces catch and fishing income variability in a highly dynamic ecosystem 

Dear Dr. Vargas:

I'm pleased to inform you that your manuscript has been deemed suitable for publication in PLOS ONE. Congratulations! Your manuscript is now with our production department. 

Kind regards, 

on behalf of

Dr. Andrea Belgrano 

Academic Editor

PLOS ONE